# Differentially Private Attention Computation

**Yeqi Gao**
The University of Washington
a916755226@gmail.com

**Zhao Song**
The Simons Institute for the Theory of Computing
at the University of California, Berkeley
magic.linuxkde@gmail.com

**Xin Yang**
The University of Washington
yangxin199207@gmail.com

**Yufa Zhou**
University of Pennsylvania
yufazhou@seas.upenn.edu

## Abstract

Large language models (LLMs), especially those based on the Transformer architecture, have had a profound impact on various aspects of daily life, such as natural language processing, content generation, research methodologies, and more. Nevertheless, a crucial concern regarding the inference results of large language models is the issue of security and privacy. Given that large language models can generate results that may leak sensitive confidential or copyright information in many scenarios, it is crucial to compute the attention matrix with provable privacy guarantees, as attention is all you need.

In this work, we propose a novel and efficient algorithm for approximating the attention matrix while providing differential privacy (DP) guarantees. To achieve this, we build on recent advancements in fast attention computation and differentially private matrix publishing.

## 1 Introduction

The development of large language models (LLMs) has been rapid and significant in recent years, with numerous breakthroughs and advancements in the field. BERT [DCLT18] achieved state-of-the-art performance on a wide range of language tasks by training on a massive amount of text data in 2018. Since then, the GPT (Generative Pre-trained Transformer) family of models has further advanced the field. GPT-2 [RWC+19] and GPT-3 [BMR+20], with billions of parameters, are able to generate highly coherent and human-like text. Other notable LLMs include XLNet [YDY+19], which addresses some of the limitations of BERT [DCLT18], and RoBERTa [LOG+19], which improves upon BERT [DCLT18]'s training methods to achieve better performance. The rapid development of LLMs has been fueled by advancements in hardware, software, and data availability, allowing researchers and companies to train and deploy these models at an unprecedented scale.

As a result of their development, LLMs have found a wide range of applications in various fields. In the field of natural language processing (NLP) [VSP+17, RNS+18, DCLT18, BMR+20], LLMs are used for tasks such as language translation [HWL21], sentiment analysis [UAS+20], and creative writing [Ope23]. In addition, LLMs are being used to develop chatbots and virtual assistants that can understand and respond to natural language queries [BMR+20, Ope23]. Outside of NLP, LLMs are being used in scientific research to generate new hypotheses and discover novel patterns in large datasets. The applications of LLMs are expanding rapidly, and it is likely that they will play an increasingly important role in many fields, such as computer vision [RF18], robotics [KNK21], and autonomous vehicles [ZTL+17, BKO18].

38th Conference on Neural Information Processing Systems (NeurIPS 2024).

Despite their many benefits, large language models (LLMs) have the potential to pose several privacy and security risks [Sag18, VKB23, KGW+23, EMM+23]. One concern is the risk of data breaches, as LLMs require large amounts of data to be trained and the data used for training is often collected from public sources without the explicit consent of the individuals involved. This data could include sensitive personal information, such as medical records, financial data, or personally identifiable information [TPG+17, ERLD17]. Furthermore, LLMs can potentially be used to generate convincing fake text [RWC+19, RSR+20], which could be used for malicious purposes such as phishing attacks, spreading misinformation or impersonating individuals online. Additionally, LLMs can be used for so-called "model inversion" attacks [FJR15], where an attacker can extract private information about individuals by querying the model. For example, an attacker could use the model to infer sensitive information, such as an individual's political views or sexual orientation, based on their text input. These privacy and security concerns highlight the need for ethical considerations and responsible use of LLMs, as well as for the development of robust security mechanisms to protect against potential attacks.

Given that attention mechanisms are at the core of models like the Transformer [VSP+17], considering privacy in attention computation is crucial. Attention mechanisms process input data that may contain sensitive information, and the computed attention weights could inadvertently reveal this information if exposed. Specifically, if sensitive data are encoded within the attention weights, compromising these weights could lead to the disclosure of personal identifying information or trade secrets.

Recent studies have focused on privacy issues related to Transformers and their attention mechanisms. For example, [VKB23] showed that learned conditional generative models might output samples similar to copyrighted data in their training set, leading to copyright infringement issues. The proposed solution is near access-freeness (NAF), which involves defining generative models that do not access potentially copyrighted data. [VKB23] provide formal definitions of NAF and generative model learning algorithms that produce models with strong bounds on the probability of sampling protected content. While NAF provides formal guarantees against such infringements, it also underscores the need to secure the attention mechanisms within Transformers to prevent privacy breaches related to sensitive information embedded in attention weights.

Moreover, the potential harms of LLMs extend to intellectual property violations and the dissemination of misinformation. To mitigate these issues, [KGW+23] developed a watermarking framework for proprietary language models. This framework embeds invisible signals into generated text that can be algorithmically detected, promoting accountability and traceability. Although this approach addresses the outputs of LLMs, it emphasizes the broader necessity of safeguarding the internal computations—like attention mechanisms—to enhance overall security.

Building upon the discussed challenges, our research focuses on addressing privacy and security issues in attention computation. Unlike previous works [ZHDK23, AS23, BSZ23, LSZ23, DLS23, GMS23, DMS23, GSY23], our work will concentrate on static computation for attention computation. To be specific, static computation is a technique used in implementing attention mechanisms in deep learning models, especially in the field of natural language processing. It involves computing the attention weights between the encoder and decoder only once and reusing them during decoding, rather than dynamically computing the attention weights for each time step during decoding. This approach enhances computational efficiency and reduces overall decoding time, especially for longer sequences, while also strengthening privacy and security in attention-based models.

## 1.1 Key Definitions

Here, let us recall the formal mathematical definition of attention computation in static setting,

**Definition 1.1** (Attention computation, see [ZHDK23, AS23, BSZ23] as examples). *Given matrices $Q \in \mathbb{R}^{n \times d}$, $K \in \mathbb{R}^{n \times d}$ and $V \in \mathbb{R}^{n \times d}$, the goal is to compute*

$$\mathsf{Att}(Q, K, V) := D^{-1}AV$$

*where $A = \exp(QK^\top) \in \mathbb{R}^{n \times n}$ (we apply $\exp()$ entry-wisely to the matrix), and $D = \mathrm{diag}(A\mathbf{1}_n)$.*

Following from the setting of work [DMS23], we consider the symmetric attention approximation problem where we treat $Q = K$ and ignore the effect of $V$. The formal formulation is

**Definition 1.2.** *Given $X \in \mathbb{R}^{n \times d}$, the goal is to find some $Y \in \mathbb{R}^{n \times m}$ such that*

$$\|D(XX^\top)^{-1} \exp(XX^\top) - D(YY^\top)^{-1} \exp(YY^\top)\| \leq \text{small}$$

*where $\| \cdot \|$ is some certain norm and $D(XX^\top) = \text{diag}(\exp(XX^\top) \cdot \mathbf{1}_n)$.*

One recent work [VKB23] choose the angle of near access-freeness to study the privacy concerns in LLMs. However, in this work, we use the differential privacy concept [DR+14], and the formal definition of differential privacy can be written as follows.

**Definition 1.3** (Differential Privacy [DMNS06, DKM+06]). *A randomized mechanism $\mathcal{M}$ is $(\epsilon, \delta)$-differentially private if for any event $\mathcal{O} \in \text{Range}(\mathcal{M})$ and for any pair of neighboring databases $S, S'$ that differ in a single data element, one has*

$$\Pr[\mathcal{M}(S) \in \mathcal{O}] \leq \exp(\epsilon) \cdot \Pr[\mathcal{M}(S') \in \mathcal{O}] + \delta.$$

Finally, we're ready to define our differentially private attention computation problem.

**Definition 1.4** (General Differentially Private Attention). *Let $f : \mathbb{R} \to \mathbb{R}$ denote some fixed function. For a given matrix $X \in \mathbb{R}^{n \times d}$ with $d \gg n$, let $\mathcal{M}$ denote some mapping that maps $\mathbb{R}^{n \times d}$ to $\mathbb{R}^{n \times n}$, let $A = \mathcal{M}(X)$, for parameter $\epsilon, \delta \in (0, 0.1)$, the goal is to design an $(\epsilon, \delta)$-differetially private algorithm that takes $X \in \mathbb{R}^{n \times d}$ as input and generates a PSD matrix $B \in \mathbb{R}^{n \times n}$ such that*

$$\| \mathsf{D}(A)^{-1} f(A) - \mathsf{D}(B)^{-1} f(B)\| \leq g(\epsilon, \delta)$$

*where $f(A)_{i,j} = f(A_{i,j})$, $\mathsf{D}(A) = \text{diag}(f(A)\mathbf{1}_n)$ and where $g$ is some function.*

Definition 1.4 is very general, and covers the standard self-attention computation. In particular, when $\mathcal{M}(X) = XX^\top$ and $f(z) = \exp(z)$, then above definition recovers the standard self-attention in LLMs.

## 1.2 Our Result

Our results rely on good properties of the input data, which are defined as follows. They play a crucial role in the analysis of sensitivity with respect to $\mathcal{M}(X) = XX^\top$ (See Section 4.3).

**Definition 1.5** (Dataset). *Fix $\eta > 0, \alpha > 0$. We say our dataset $X \in \mathbb{R}^{n \times d}$ is $(\alpha, \eta)$-good if $XX^\top \succeq \eta \cdot I_n$ and for all $i \in [d]$, $\|X_{*,i}\|_2 \leq \alpha$.*

In addition, we will introduce our proposed definition of neighboring data as follows.

**Definition 1.6** (Neighboring data). *Let $X, \widetilde{X} \in \mathbb{R}^{n \times d}$ denote two datasets from distribution $\mathcal{D}$, we say that $X$ and $\widetilde{X}$ are $\beta$-close if there exists exact one $i \in [d]$ so that $\|X_{*,i} - \widetilde{X}_{*,i}\|_2 \leq \beta$ and for all $j \in [d] \backslash \{i\}$, $X_{*,j} = \widetilde{X}_{*,j}$. In this work, we consider two datasets to be neighboring if they are $\beta$-close.*

The above definition facilitates a more straightforward analysis of the sensitivity of attention matrix computations. By regulating $\beta$-closeness, we can establish bounds on how the attention matrix responds to minor variations in input data, which is essential for ensuring differential privacy guarantees. Furthermore, in practical scenarios, assessing dataset similarity based on feature-wise differences rather than individual data points can be more practical and aligns better with real-world considerations.

Based on the aforementioned definitions, our work demonstrates the sensitivity property of $\mathcal{M}(X) = XX^\top$ (attention matrix computation). Furthermore, we present a novel and efficient algorithm for approximating the attention matrix, which combines error analysis on matrix perturbation with provable privacy guarantees. We state our result as follows:

**Theorem 1.7** (Main result, informal of Theorem 3.1). *Let $d \geq n$. Let $X \in \mathbb{R}^{n \times d}$. Let $f(z) \in \{\exp(z), \cosh(z)\}$. Let $r, \epsilon, \delta \in (0, 0.1)$. Let $\Delta = 0.1 \min\{\frac{\epsilon}{\sqrt{k \log(1/\delta)}}, \frac{\epsilon}{\log(1/\delta)}\}$. Let $A = \mathcal{M}(X) = XX^\top$ and $\|A\|_\infty \leq r$. Let $f(A)$ and $\mathsf{D}(A)$ be defined as Definition 1.4. For all $X$ sampled from $\mathcal{D}$, $X$ is $(\alpha, \eta)$-good (see Definition 1.5). Let $\eta < r$. Let $\beta$ be the parameter for the neighboring dataset. Let $2\alpha\beta\sqrt{n}/\eta < \Delta$. Suppose $\|\mathcal{M}(X)^{1/2}\mathcal{M}(\widetilde{X})^{-1}\mathcal{M}(X)^{1/2} - I\|_F \leq \Delta$*

*for all* $X \in \mathbb{R}^{n \times d}, \widetilde{X} \in \mathbb{R}^{n \times d}$ *(see Definition 1.6). Let* $\rho = \sqrt{(n^2 + \log(1/\gamma))/k} + (n^2 + \log(1/\gamma))/k < 0.1\epsilon$. *Then, there is an algorithm (Algorithm 1) that takes* $X$ *as input and produces the matrix* $B \in \mathbb{R}^{n \times n}$ *and also general attention* $\mathsf{D}(B)^{-1}f(B)$ *as output such that*

$$\| \mathsf{D}(A)^{-1}f(A) - \mathsf{D}(B)^{-1}f(B) \|_\infty \leq 4 \cdot (1 + \epsilon + 2r) \cdot r$$

*which holds with probability* $1 - \gamma$. *With respect to* $X$, *the algorithm is* $(\epsilon, \delta)$-*differential private.*

## 1.3 Related Work

**Differential Privacy and Deep Learning.** Differential privacy (DP) is a rigorous and quantifiable notion of privacy that ensures individual data entries cannot be distinguished within a dataset. It has become the go-to standard for understanding information leakage [DR+14]. This widely recognized framework is increasingly being adopted in industry and has many real-world applications [XZA+23, TM22, Sna22, Fac22, Dif22, RSY+21]. There has been extensive research on applying differential privacy in deep learning [ACG+16, KKM+20, GGK+21, LSSZ24a, SYYZ23, LSSS24, LLS+24b]. Recent works [YNB+21, LTLH21] have applied DP-SGD [ACG+16] to large language models (LLMs) for private fine-tuning. Our research, however, is orthogonal to these works as we focus on attention computation and consider general differential privacy mechanisms, not just DP-SGD.

**Roadmap.** Our paper is organized as follows. Section 2 presents the notations that are used throughout our paper. Our main result is presented in Section 3. We provide an overview of our techniques in Section 4. In Section 5, we give our conclusion of the paper.

## 2 Notations

$\mathbb{E}[X]$ represents the expected value (or mean) of a random variable $X$. We use $\chi_d^2$ to denote a Chi-squared random variable with $d$ degrees of freedom. If $M$ and $N$ are symmetric matrices, we define $M \succeq N$ to mean that for all vectors $x$, the inequality $x^\top M x \geq x^\top N x$ holds. If $M$ is a symmetric matrix of dimension $n \times n$, we define $M$ to be positive semidefinite ($M \succeq 0$) if the inequality $x^\top M x \geq 0$ holds for all vectors $x \in \mathbb{R}^n$. We use the notation $\mathbf{0}_n$ to denote an $n$-dimensional vector whose entries are all zero, and $\mathbf{1}_n$ to denote an $n$-dimensional vector whose entries are all one. The symbol $I_n$ represents the $n \times n$ identity matrix, which is a square matrix with ones on the main diagonal and zeros elsewhere. Let $x$ be an arbitrary vector in $\mathbb{R}^n$. We define $\exp(x) \in \mathbb{R}^n$ as a vector whose $i$-th entry $\exp(x)_i$ is equal to $\exp(x_i)$, where $\exp(\cdot)$ denotes the exponential function. We use $\langle x, y \rangle$ to denote $\sum_{i=1}^n x_i y_i$. For any matrix $A$, we use $\|A\|$ to denote the spectral norm of $A$, i.e., $\|A\| = \max_{\|x\|_2 = 1} \|Ax\|_2$, $\|A\|_F$ to denote its Frobenius norm and $\|A\|_\infty$ to denote the infinity norm. $A_{i,j}$ represents the element in the $i$-th row and $j$-th column of matrix $A$.

## 3 Main Result

---
**Algorithm 1** Differential privacy algorithm
---
1: **procedure** DPATTENTION($X$)
2:     $A \leftarrow XX^\top$
3:     $B \leftarrow$ DPCOVARIANCE($A, k$)                    ▷ See Algorithm 2.
4:     Compute $f(B)$
5:     Compute $\mathsf{D}(B)^{-1}f(B)$
6: **end procedure**
---

In this section, we provide a theoretical analysis of Algorithm 1, our primary algorithm for differentially private general attention computation. Our analysis leverages the tools established in Section 4.1, Section 4.2, and Section 4.3. From our previous proofs, it is evident that our algorithm possesses a rigorous differential privacy property, offering new insights into both differential privacy and attention mechanisms.

**Theorem 3.1** (Main result). *If all of the following requirements are met Let* $d \geq n$, $X \in \mathbb{R}^{n \times d}$, *and* $f(z) \in \{\exp(z), \cosh(z)\}$. *We define* $r \in (0, 0.1)$ *as bounded ratio and* $\epsilon, \delta \in (0, 0.1)$ *as*

the parameter of DP. Let $\Delta = 0.1 \min\{\frac{\epsilon}{\sqrt{k \log(1/\delta)}}, \frac{\epsilon}{\log(1/\delta)}\}$. Let $A = \mathcal{M}(X) = XX^\top$ and $\|A\|_\infty \leq r$. For all $X$ sampled from $\mathcal{D}$, $X$ is $(\alpha, \eta)$-good (see Definition 1.5). Let $\eta < r$. Let $\beta$ be the parameter for neighboring dataset. Let $2\alpha\beta\sqrt{n}/\eta < \Delta$. Let $\Delta$ denote the sensitivity parameter that $\mathcal{M}$ satisfies a sensitivity bound that $\|\mathcal{M}(X)^{1/2}\mathcal{M}(\widetilde{X})^{-1}\mathcal{M}(X)^{1/2} - I\|_F \leq \Delta$ for any neighboring datasets $X \in \mathbb{R}^{n \times d}, \widetilde{X} \in \mathbb{R}^{n \times d}$ (see Definition 1.6). Let $\rho = \sqrt{(n^2 + \log(1/\gamma))/k} + (n^2 + \log(1/\gamma))/k$ and $\rho < 0.1\epsilon$.

There is an algorithm (Algorithm 1) that takes $X$ as input and produces the matrix $B \in \mathbb{R}^{n \times n}$ and also attention $\mathsf{D}(B)^{-1}f(B)$ as output such that

- **Part 1.** $\|\mathsf{D}(A)^{-1}f(A) - \mathsf{D}(B)^{-1}f(B)\|_\infty \leq 4 \cdot (1 + \epsilon + 2r) \cdot r$.

- **Part 2.** *With respect to $X$, the algorithm is $(\epsilon, \delta)$-differential private.*

- **Part 3.** *It holds with probability $1 - \gamma$.*

*Proof of Theorem 3.1.* The proof can be divided into two parts as follows.

**Proof of Part 1 and Part 3.** Our proof focus on the function $\mathcal{M}(X) := XX^\top$ first. Let $\alpha$ and $\eta$ be denoted in Definition 1.5 and $\beta$ be denoted as Definition 1.6. Based on the assumption on dataset above, we can obtain $X$ is $(\eta, \alpha)$-good (See Definition 1.5) while $X$ and $\widetilde{X}$ are $\beta$-close (See Definition 1.6).

According to **Part 1** of Lemma 4.11, we can conclude the property on $\mathcal{M}(X) = XX^\top$ such that

$$\|(XX^\top)^{-1/2}\widetilde{X}\widetilde{X}^\top(XX^\top)^{-1/2} - I\|_F \leq 2\sqrt{n}\alpha\beta/\eta$$

Let $\mathcal{M}$ be the function denoted in the theorem statement and let $\rho$ be denoted as follows:

$$\rho := O(\sqrt{(n^2 + \log(1/\gamma))/k} + (n^2 + \log(1/\gamma))/k)$$

Now, we will apply the conclusion drawn in Section 4.2. In order to satisfy the requirement specified in **Requirement 4** of Theorem 4.9, we need $\mathcal{M}(X)$ to meet the following assumption:

$$\|\mathcal{M}(X)^{1/2}\mathcal{M}(\widetilde{X})^{-1}\mathcal{M}(X)^{1/2} - I\|_F \leq \Delta.$$

Now, if we choose $2\alpha\beta\sqrt{n}/\eta < \Delta$. we will guarantee that our $\mathcal{M}(X)$ satisfies the assumption specified in **Requirement 4** of Theorem 4.3. According to **Part 3** of Theorem 4.3, there exists Algorithm 2 which can produce a matrix $B \in \mathbb{R}^{n \times n}$ such that, with probability at least $1 - \gamma$

$$(1 - \rho)A \preceq B \preceq (1 + \rho)A \tag{1}$$

By choosing $\rho \in (0, 0.1)\epsilon$, we will have

$$(1 - \epsilon)B \preceq A \preceq (1 + \epsilon)B \tag{2}$$

Now according to Theorem 4.3 and Eq. (2), we have

$$\|\mathsf{D}(A)^{-1}f(A) - \mathsf{D}(B)^{-1}f(B)\|_\infty \leq 4 \cdot (1 + \epsilon + 2r) \cdot r$$

Now, the proofs of **Part 1** and **Part 3** are completed.

**Proof of Part 2.** It simply follows from **Part 1** of Theorem 4.9. □

The main result implies that we can design an algorithm that computes a private approximation of the attention mechanism used in neural networks for functions like $f(z) = \exp(z)$ or $f(z) = \cosh(z)$. Specifically, under certain conditions on the input matrix $X$ and parameters $\epsilon, \delta$, and with a small bounded ratio $r$, the algorithm produces a matrix $B$ such that the normalized attention matrices derived from $A = XX^\top$ and $B$ are close in the infinity norm. This closeness is quantified by a bound proportional to $r$, ensuring that the utility of the attention mechanism is preserved. Additionally, the algorithm is $(\epsilon, \delta)$-differentially private with respect to $X$, meaning it protects individual data entries from being inferred. The privacy and utility guarantees hold with high probability $1 - \gamma$, demonstrating that it is possible to implement attention mechanisms in a way that maintains both model performance and data privacy.

# 4 Technique Overview

The objective of our research is to develop a differentially private algorithm that addresses the challenges of computing attention on large datasets. Specifically, we focus on scenarios where the size of the data matrix $X$ is extremely large, with the number of features $d$ significantly exceeding the number of samples $n$ (i.e., $d \gg n$). In these cases, the attention matrix $A$ is obtained as the output of the function $\mathcal{M}(X) = XX^\top$, and our goal is to ensure that the computation of $A$ is performed in a differentially private [DMNS06, DKM$^+$06] manner.

**Perturb PSD Matrix.** We define the attention computation $\mathsf{D}(X)$ as Definition 4.2. By employing a more general version of Perturbation analysis presented in [DMS23], we select $f$ as specified in Definition 4.1. To complete the error analysis of attention computation, we will utilize the perturbation analysis of the diagonal normalization matrix and the PSD matrix presented in Appendix C. Under the assumption the relative error between input matrix $\mathcal{M}(X) := A$ and privacy required matrix output $B$ is less than or equal to $\epsilon \in (0, 0.1)$ where $(1 - \epsilon)B \preceq A \preceq (1 + \epsilon)B$. To establish an upper bound for $\| \mathsf{D}(A)^{-1} f(A) - \mathsf{D}(B)^{-1} f(B) \|_\infty$, we first derive the following bound:

- **Part 1.** $| \mathsf{D}(A)_{i,i} - \mathsf{D}(B)_{i,i} | \leq c_1 \cdot r \cdot \min\{\mathsf{D}(A)_{i,i}, \mathsf{D}(B)_{i,i}\} \ \forall i \in [n]$,
- **Part 2.** $|f(A_{i,j}) - f(B_{i,j})| \leq c_2 \cdot r \cdot \min\{f(A_{i,j}), f(B_{i,j})\} \ \forall i, j \in [n] \times [n]$

And with the error of attention computation under control as mentioned above, we can obtain:

$$\| \mathsf{D}(A)^{-1} f(A) - \mathsf{D}(B)^{-1} f(B) \|_\infty \leq 4 \cdot (1 + \epsilon + 2r) \cdot r$$

**Sensitivity for PSD Matrix.** Our work relies on the basic assumptions that $X \in \mathbb{R}^{n \times d}$ is a $(\eta, \alpha)$-good dataset (See Definition 1.5) and that $X$ and $\widetilde{X}$ are $\beta$-close to each other (See Definition 1.6). We choose $\mathcal{M}(X) := XX^\top$. Now we will demonstrate the property of our function $\mathcal{M}(X) = XX^\top$ based on the given assumptions. Since $X$ and $\widetilde{X}$ are neighbor datasets, we have the following:

$$\|\mathcal{M}(X)^{1/2} \mathcal{M}(\widetilde{X})^{-1} \mathcal{M}(X)^{1/2} - I\|_F \leq 2\alpha\beta\sqrt{n}$$

The proof details can be found in Section E. Let us denote $\Delta$ as defined in Definition 4.5. By choosing $2\alpha\beta\sqrt{n}/\eta < \Delta$, we will have

$$\|( \underbrace{XX^\top}_{:= \mathcal{M}(X)} )^{1/2} ( \underbrace{\widetilde{X}\widetilde{X}^\top}_{:= \mathcal{M}(\widetilde{X})} )^{-1} ( \underbrace{XX^\top}_{:= \mathcal{M}(X)} )^{1/2} - I\|_F \leq \Delta \tag{3}$$

The assumption specified in the **Requirement 5** of Theorem D.7 will be satisfied. Next, we will introduce our main algorithm using Eq. (3).

**Differential Privacy Algorithm.** Next we give the differential privacy algorithm described in Theorem 1.7. And we will demonstrate that our algorithm (Algorithm 1) is able to output a matrix that satisfies the **Part 1** of our formal main result (See Theorem 3.1).

To begin with, we demonstrate that there exists an algorithm capable of taking input $A$ and producing a matrix $B$ as output such that the difference between $A$ and $B$ is small enough, which can be seen as a small error resulting from the perturbation of $A$ by $\rho := O(\sqrt{(n^2 + \log(1/\gamma))/k} + (n^2 + \log(1/\gamma))/k)$. In other words, we have $(1 - \rho)A \preceq B \preceq (1 + \rho)A$. The above equation holds with probability $1 - \gamma$. Note that $k$ and $\gamma$ can be chosen according to our requirements. We can ensure that a satisfactory $\rho$ is obtained. By choosing a small enough $\rho \leq 0.1\epsilon$ and using the conclusions on perturbed PSD matrices, the algorithm can certainly output a satisfactory $B$ which promises our attention computation is privacy [DMNS06, DKM$^+$06].

## 4.1 Error Control from Logit Matrix to Attention Matrix

In this section, we analyze the perturbations in the attention computation, which are used to control the error. First, we define the followings.

**Definition 4.1.** *Let $f(z)$ denote one of the following functions* $\exp(z)$ *and* $\cosh(z)$.

The motivation of considering $\exp(z)$ is due to recent LLMs. The motivation of considering $\cosh(z)$ is from recent progress in potential function design of convex optimization [CLS19, LSZ19, Son19, Bra20, JSWZ21, GS22, QSZZ23].

**Definition 4.2.** *Given that $A \in \mathbb{R}^{n \times n}$, we define $f$ as Definition 4.1. Let us define $\mathsf{D}(A) := \mathrm{diag}(f(A)\mathbf{1}_n)$ where we apply $f$ to matrix entrywisely.*

We state a major tool we proved in this paper to control the error propagation which summarizes the effectiveness of our error control mechanisms in achieving differential privacy for the computation of the attention matrix.

**Theorem 4.3.** *Let $\epsilon \in (0, 0.1)$ and $r \in (0, 0.1)$. Let $\|A\|_\infty \leq r$ and $(1 - \epsilon)B \preceq A \preceq (1 + \epsilon)B$. We define $\mathsf{D}$ as Definition 4.2 and $f$ as Definition 4.1. Then, we have*
$$\| \mathsf{D}(A)^{-1}f(A) - \mathsf{D}(B)^{-1}f(B)\|_\infty \leq 4 \cdot (1 + \epsilon + 2r) \cdot r.$$

The prior work [DMS23] only work for $\exp()$ function and the final guarantee is $O(r)$. We generalize it to $\cosh()$ function also, and our error bound is much tighter. The proof of the theorem above is in Section C.4.

## 4.2 Analysis of Gaussian Sampling Mechanism

This section introduces a crucial component of our main differential privacy algorithm (Algorithm 1): the differentially private covariance releasing mechanism, detailed in Algorithm 2. The differential privacy (DP) of Algorithm 2 ensures the DP of the main algorithm (Algorithm 1). Therefore, we will also demonstrate its DP. Our proof is based on the assumption that the sensitivity is bounded (Requirement 4 in Theorem 4.9). We defer the validation of the assumption to Section 4.3. For clarity, the following proof is based on the assumption that $M \leq \Delta$ (See Definition 4.4 and Definition 4.5), which will be proven in Section 4.3. Let $\mathcal{Y}$ and $\mathcal{Y}'$ be neighboring datasets, as denoted in Definition 1.6.

To facilitate the explanation in the following proof, we will define $M$ to better illustrate the properties of $\mathcal{M}$.

**Definition 4.4.** *Let $\mathcal{M} : (\mathbb{R}^n)^d \rightarrow \mathbb{R}^{n \times n}$ be a (randomized) algorithm that given a dataset of $d$ points in $\mathbb{R}^n$ outputs a PSD matrix. Then, we define $M := \sup_{\mathcal{Y}, \mathcal{Y}'} \|\mathcal{M}(\mathcal{Y})^{1/2}\mathcal{M}(\mathcal{Y}')^{-1}\mathcal{M}(\mathcal{Y})^{1/2} - I\|_F$. Here $\sup$ is over all neighboring datasets $\mathcal{Y}$ and $\mathcal{Y}'$ (see Definition 1.6).*

We define the upper bound of $M$ as $\Delta$ as follows.

**Definition 4.5.** *Let $M$ be defined in Definition 4.4. We define $\Delta := \min \left\{ \frac{\epsilon}{\sqrt{8k\log(1/\delta)}}, \frac{\epsilon}{8\log(1/\delta)} \right\}$ such that $M \leq \Delta$.*

We define the notation of gaussian distribution below.

**Definition 4.6** (Gaussian Distribution). *We denote the $\mathcal{N}(0, \Sigma)$ density function as follows $f_\Sigma(x) = (2\pi)^{-\frac{n}{2}} \det(\Sigma)^{-\frac{1}{2}} \exp(-0.5x^\top \Sigma x)$.*

We state our differentially private covariance releasing algorithm. Assuming $g_i$s are i.i.d. samples from $f_\Sigma(x)$ for $i \in [k]$, we use $g_1, g_2, \cdots, g_k$ to compute the covariance estimate $\widehat{\Sigma}$ in Algorithm 2. We will demonstrate the analysis of $\widehat{\Sigma} = \frac{1}{k}\sum_{i=1}^k g_i g_i^\top$ using the symbol from Appendix D, which leads to the privacy guarantee for our algorithm 2.

---

**Algorithm 2** Differentially private covariance releasing

1: **procedure** DPCOVARIANCE($\Sigma \in \mathbb{R}^{n \times n}, k \in \mathbb{N}$)          $\triangleright$ PSD matrix $\Sigma$ and parameter $k$
2:     Obtain vectors $g_1, g_2, \cdots, g_k$ by sampling $g_i \sim \mathcal{N}(0, \Sigma)$, independently for each $i \in [k]$
3:     Compute $\widehat{\Sigma} = \frac{1}{k}\sum_{i=1}^k g_i g_i^\top$          $\triangleright$ This is Covariance estimate.
4:     **return** $\widehat{\Sigma}$
5: **end procedure**

---

The soundness Algorithm 2 can be shown using Theorem 4.9. We now give the definitions of $\Sigma_1, \Sigma_2, h_{i,j}$ and $Z$ which will be used to prove the Theorem 4.9.

**Definition 4.7.** *Let $\mathcal{M}$ be denoted in Definition 4.4 and $\Sigma(\mathcal{Y}) := \mathcal{M}(\mathcal{Y})$. We define $\Sigma_1 := \Sigma(\mathcal{Y})$, $\Sigma_2 := \Sigma(\mathcal{Y}')$.*

**Definition 4.8.** *Let $g_1, g_2, \cdots, g_k$ be i.i.d samples from $\mathcal{N}(0, \Sigma_1)$ output by Algorithm 2. Then, we define $h_{i,j} := \langle \Sigma_1^{-1/2} g_i, v_j \rangle$, $Z := \sum_{i=1}^{k} \log(\frac{f_{\Sigma_1}(g_i)}{f_{\Sigma_2}(g_i)})$ where $\Sigma_1, \Sigma_2$ are defined by Definition 4.7. Note that the random variables $h_{i,j}$ are i.i.d copies of $\mathcal{N}(0,1)$.*

We will now present our theorem for Algorithm 2. The proof is delayed to Section D.6.

**Theorem 4.9** (Informal version of Theorem D.7)**.** *If all of the following requirements are met:* **Requirement 1.** *Let $\epsilon \in (0,1)$ and $\delta \in (0,1)$.* **Requirement 2.** *$k \in \mathbb{N}$.* **Requirement 3.** *Let $\Delta$ be denoted as Definition 4.5 and $\Delta < 1$.* **Requirement 4.** *Let $M, \mathcal{M}$ be denoted as Definition 4.4 and $M \leq \Delta$.* **Requirement 5.** *An input $\Sigma = \mathcal{M}(\mathcal{Y})$.* **Requirement 6.** *$\rho = O(\sqrt{(n^2 + \log(1/\gamma))/k} + (n^2 + \log(1/\gamma))/k)$.*

*Then, there is an algorithm (Algorithm 2) such that*

- *Part 1. Algorithm 2 is $(\epsilon, \delta)$-DP (with respect to the original dataset $\mathcal{Y}$).*

- *Part 2. outputs $\widehat{\Sigma}$ such that with probability at least $1 - \gamma$, $\|\Sigma^{-1/2}\widehat{\Sigma}\Sigma^{-1/2} - I_n\|_F \leq \rho$.*

- *Part 3. $(1 - \rho)\Sigma \preceq \widehat{\Sigma} \preceq (1 + \rho)\Sigma$.*

Using this theorem, we can see our Algorithm 2 is DP, which ensures the DP of Algorithm 1.

### 4.3 Sensitivity for PSD Matrix

We have demonstrated the existence of a differential privacy algorithm under the assumption on $\mathcal{M}(X) = XX^\top$ introduced in Section 4.2. In this section, we show that $\mathcal{M}(X) = XX^\top$ satisfies the assumption specified in **Requirement 4** of Theorem 4.9 for $\mathcal{M}(X)$. The lemma following is based on the assumption on datasets $X, \widetilde{X}$ (See Definition 1.5 and Definition 1.6).

**Lemma 4.10** (Informal version of Lemma E.1)**.** *If $X \in \mathbb{R}^{n \times d}$ and $\widetilde{X} \in \mathbb{R}^{n \times d}$ are neighboring dataset (see Definition 1.5 and Definition 1.6), then $(1 - 2\alpha\beta/\eta)XX^\top \preceq \widetilde{X}\widetilde{X}^\top \preceq (1 + 2\alpha\beta/\eta)XX^\top$.*

Now, we can have the following lemma. The subsequent lemma can be viewed as a variation of Lemma 4.10, yet it presents a more apparent result that can be directly employed in subsequent analyses.

**Lemma 4.11.** *Let $\alpha$ and $\eta$ be defined in Definition 1.5. Let $\beta$ be defined in Definition 1.6. Let $X$ and $\widetilde{X}$ be neighboring datasets such that $(1 - 2\alpha\beta/\eta)XX^\top \preceq \widetilde{X}\widetilde{X}^\top \preceq (1 + 2\alpha\beta/\eta)XX^\top$. Then, we have $\|(XX^\top)^{-1/2}\widetilde{X}\widetilde{X}^\top(XX^\top)^{-1/2} - I\| \leq 2\alpha\beta/\eta$ and $\|(XX^\top)^{-1/2}\widetilde{X}\widetilde{X}^\top(XX^\top)^{-1/2} - I\|_F \leq 2\sqrt{n}\alpha\beta/\eta$.*

The proof of Lemma 4.11 follows directly from the Lemma 4.10.

Presently, we have delved into the sensitivity property of attention computation. We are able to illustrate that the computation of the attention matrix aligns with the assumptions introduced in Section 4.2. Building upon this foundation, we will subsequently address our primary result in Section 3.

## 5 Conclusion

In this work, we propose a differentially private algorithm for approximating the attention matrix. Our algorithm is built upon recent advances in fast attention computation and private matrix releasing. To the best of our knowledge, this is the first work of accelerating attention computation in the DP setting. Given the dominating presence of Transformer based language models, we hope our work can stand as a starting point for fully DP training and inferring algorithms on large language models. It is also an interesting open problem to approximate asymmetric attention computation with differential privacy.

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

# Appendix

## Contents

**Roadmap.**    Section A provides more related work. In Section B, we present preliminaries for the paper. In Section C, we analyze the perturbations in attention computation. Section D presents the proof of the existence of differential privacy using our Gaussian sampling mechanism. In Section E, we provide more Lemma about sensitivity.

# A    More Related Work

**Attention Mechanism.**    Attention mechanisms are foundational in modern neural networks, gaining widespread adoption since their introduction in [VSP+17]. They are crucial in decoder-only LLMs [RWC+19] and the Vision Transformer (ViT) [DBK+20], driving significant progress in language models and computer vision [RBL+22, WSD+23, WXZ+24, WCZ+23, SWXL24, LSSY24]. Additionally, attention mechanisms have been applied to multi-modal models [AS24b, XGH+21, ZHJL24, LSSZ24b, WMS+24], mathematical reasoning [LLS+24a, XSL24], diffusion models [PX23, LSSZ24c, HWSL24], differential privacy [BEPP22, SSC+22, LSSZ24a, CSY23], Hopfield models [HYW+23, WHL+24, HLSL24, XHH+24, WHHL24, HCL+24, HCW+24], and various other techniques [LSS+24a, LSY24, QSZZ23, LSS+24b, LLS+24c, LLS+24d, CLS+24, SMN+24, LLSS24, XSW+23].

**Softmax Computation and Regression.**    With the rapid development of large language models and attention schemes, many works have focused on softmax computation and regression in this field. [AS23, AS24a] shows that a faster attention algorithm can be designed by leveraging the matrix implicitly. [BSZ23] proposes a more efficient algorithm for computing dynamic attention by employing the method of lazy update. To solve $\exp$, $\cosh$, and $\sinh$ regressions with input sparsity, [LSZ23] use an approximate Newton method that operates in near-linear time. In their work on softmax regression, [DLS23] conduct a further analysis of attention schemes based on prior research in regression. In contrast, [GMS23] focus on the convergence analysis of overparameterized two-layer networks with exponential activation functions. To compute the attention matrix more efficiently for large feature dimensions, [DMS23] propose a randomized algorithm.

# B    Preminary

Section B.1 presents the notations that are used throughout our paper. These notations are essential for a clear and concise presentation of our work. In Section B.2, we provide an introduction to some basic algebraic concepts that are relevant to our research. This includes fundamental mathematical operations and properties that are used in the analysis and development of our differential privacy algorithm. In Section B.3, we provide the previous tools that help our proofs.

## B.1    Notations

For a event $C$, $\Pr[C]$ represents the probability of event $C$ occurring. $\mathbb{E}[X]$ represents the expected value (or mean) of a random variable $X$.

We use $\chi_d^2$ to denote a Chi-squared random variable with $d$ degrees of freedom. $\mathbb{N}$ represents the set of natural numbers, which consists of all positive integers including 1, 2, 3, and so on.

If $M$ and $N$ are symmetric matrices, we define $M \succeq N$ to mean that for all vectors $x$, the inequality $x^\top M x \geq x^\top N x$ holds.

If $M$ is a symmetric matrix of dimension $n \times n$, we define $M$ to be positive semidefinite ($M \succeq 0$) if the inequality $x^\top M x \geq 0$ holds for all vectors $x \in \mathbb{R}^n$.

We use the notation $\mathbf{0}_n$ to denote an $n$-dimensional vector whose entries are all zero, and $\mathbf{1}_n$ to denote an $n$-dimensional vector whose entries are all one. The symbol $I_n$ represents the $n \times n$ identity matrix, which is a square matrix with ones on the main diagonal and zeros elsewhere.

Let $x$ be an arbitrary vector in $\mathbb{R}^n$. We define $\exp(x) \in \mathbb{R}^n$ as a vector whose $i$-th entry $\exp(x)_i$ is equal to $\exp(x_i)$, where $\exp(\cdot)$ denotes the exponential function. We use $\langle x, y \rangle$ to denote $\sum_{i=1}^n x_i y_i$.

For any matrix $A$, we use $\|A\|$ to denote the spectral norm of $A$, i.e., $\|A\| = \max_{\|x\|_2=1} \|Ax\|_2$, $\|A\|_F$ to denote its Frobenius norm and $\|A\|_\infty$ to denote the infinity norm. $A_{i,j}$ represents the element in the $i$-th row and $j$-th column of matrix $A$. $\det(A)$ represents the determinant of matrix

$A$. For a square and symmetric matrix $A \in \mathbb{R}^{n \times n}$, we say $A$ positive semi-definite ($A \succeq 0$) if for all vectors $x \in \mathbb{R}^n$, we have $x^\top A x \geq 0$.

We denote the inverse of a matrix $M$ as $M^{-1}$ and its transpose as $M^\top$. We refer to $\lambda_i$ as the $i$-th eigenvalue of $N$.

$\mathbb{S}_+^n$ denotes the set of $n \times n$ positive semidefinite (PSD) matrices.

### B.2 Basic Algebra

In this section, we offer an introduction to fundamental algebraic concepts.

**Fact B.1.** *We have*

- *Part 1. Let $A \in \mathbb{R}^{n \times n}$, then we have $\|A\|_F \leq \sqrt{n}\|A\|$.*

- *Part 2. Let $A \in \mathbb{R}^{n \times n}$, then we have $\|A\| \leq \|A\|_F$*

- *Part 3. For two vectors $a, b \in \mathbb{R}^n$, then we have $\|ab^\top\| \leq \|a\|_2 \cdot \|b\|_2$*

**Fact B.2.** *We have*

- *Part 1. $\cosh(x) = \sum_{i=0}^{\infty}(1/(2i)!) \cdot x^{2i}$.*

- *Part 2. $\exp(x) = \sum_{i=0}^{\infty}(1/(i!)) \cdot x^i$.*

- *Part 3. We have $|\exp(x) - 1| \leq |x| + x^2$, $\forall x \in (-0.1, 0.1)$.*

- *Part 4. $|\exp(x) - \exp(y)| \leq \exp(x) \cdot (|x - y| + |x - y|^2)$ for $|x - y| \leq 0.1$.*

- *Part 5. We have $|\cosh(x) - 1| \leq x^2$, $\forall x \in (-0.1, 0.1)$.*

- *Part 6. $|\cosh(x) - \cosh(y)| \leq \cosh(x) \cdot |x - y|^2$ for $|x - y| \leq 0.1$.*

### B.3 Previous Tools

This section introduces several differential privacy tools. These tools are essential for demonstrating the differential privacy properties of our algorithm.

**Theorem B.3** (Empirical covariance estimator for Gaussian [Ver18]). *Let $\Sigma \in \mathbb{R}^{d \times d}$ be PSD, $X_1, \cdots, X_n \sim \mathcal{N}(0, \Sigma)$ be i.i.d and $\widetilde{\Sigma} = \frac{1}{n}\sum_{i=1}^{n} X_i X_i^\top$. Then with probability $1 - \gamma$, it holds that $\|\Sigma^{-1/2}\widetilde{\Sigma}\Sigma^{-1/2} - I\|_F \leq \rho$ for some $\rho = O(\sqrt{\frac{d^2 + \log(1/\gamma)}{n}} + \frac{d^2 + \log(1/\gamma)}{n})$.*

**Theorem B.4** (Lemma 1.5 in [Vad17], Section 1.1 of [BS16]). *For a (randomized) mechanism $\mathcal{M}$ and datasets $x, y$, define the function $f_{xy}(z) := \log(\frac{\Pr[\mathcal{M}(x)=z]}{\Pr[\mathcal{M}(y)=z]})$ If $\Pr[f_{xy}(\mathcal{M}(x)) > \epsilon] \leq \delta$ for all adjacent datasets $x, y$, then $\mathcal{M}$ is $(\epsilon, \delta)$-DP.*

**Lemma B.5** (Sub-exponential tail bound, Proposition 2.9 in [Wai19]). *Suppose that $X$ is sub-exponential with parameters $(\nu, \alpha)$. Then $\Pr[X - \mu \geq t] \leq \max\{\exp(-\frac{t^2}{2\nu^2}), \exp(\frac{t}{2\alpha})\}$.*

**Lemma B.6** ($\chi_1^2$ sub-exponential parameters, Example 2.11 in [Wai19]). *A chi-squared random variable with 1 degree of freedom $(\chi_1^2)$ is sub-exponential with parameters $(\nu, \alpha) = (2, 4)$*

**Lemma B.7** (Sub-exponential parameters of independent sum, Chapter 2 of [Wai19]). *Consider an independent sequence $X_1, \cdots, X_k$ of random variables, such that $X_i$ is sub-exponential with parameters $(\nu_i, \alpha_i)$. Then the variable $\sum_{i=1}^{k} X_i$ is sub-exponential with parameters $(\nu_*, \alpha_*)$, where $a_* = \max_{i \in [k]} \alpha_i$ and $\nu_* = (\sum_{i=1}^{k} \nu_i^2)^{1/2}$.*

## C Error Control from Logit Matrix to Attention Matrix

In Section C.1, we discuss the perturbation of positive semi-definite (psd) matrices, which is a crucial step in ensuring the differential privacy of our algorithm. Section C.2 focuses on the perturbation of diagonal normalization matrices, which is another important aspect of our error control approach. In Section C.3, we analyze the error in the attention matrix computation that arises from these

perturbations. Finally, in Section C.4, we present the main result of Section C, which summarizes the effectiveness of our error control mechanisms in achieving differential privacy for the computation of the attention matrix.

## C.1 Perturb PSD Matrix

In Section C.1, we discuss the perturbation of positive semi-definite (psd) matrices. This is a crucial step in ensuring the differential privacy of our algorithm.

**Lemma C.1** (Lemma 3.1 in [DMS23]). *We denote $A \in \mathbb{R}^{n \times n}$ and $B \in \mathbb{R}^{n \times n}$ as psd matrices.*

*If all of the following requirements are met*

- **Requirement 1.** *We have $-r \leq A_{i,j} \leq r$, $\forall (i,j) \in [n] \times [n]$.*

- **Requirement 2.** $(1 - \epsilon)B \preceq A \preceq (1 + \epsilon)B$;

*Then, it follows that*

$$B_{i,j} \in [-(1 + \epsilon)r, (1 + \epsilon)r].$$

**Lemma C.2** (A general version of Lemma 3.2 in [DMS23]). *If all of the following requirements are met*

- **Requirement 1.** $A_{i,j} \in [-r, r]$.

- **Requirement 2.** $B_{i,j} \in [-(1 + \epsilon)r, (1 + \epsilon)r]$.

- **Requirement 3.** $r \in (0, 0.1)$, $\epsilon \in (0, 0.1)$.

- **Requirement 4.** *Let $f(z) \in \{\exp(z), \cosh(z)\}$.*

*It follows that*

- **Part 1.**

$$|f(A_{i,j}) - f(B_{i,j})| \leq f(A_{i,j}) \cdot (2 + 2\epsilon + 4r) \cdot r \ \ \forall i, j \in [n] \times [n].$$

- **Part 2.**

$$|f(A_{i,j}) - f(B_{i,j})| \leq f(B_{i,j}) \cdot (2 + 2\epsilon + 4r) \cdot r \ \ \forall i, j \in [n] \times [n].$$

*Proof.* According to **Requirement 1.**, **Requirement 2.** and **Requirement 3.**, we have

$$|A_{i,j} - B_{i,j}| \leq (2 + \epsilon)r. \tag{4}$$

**Proof of Part 1.** It follows that

$$\begin{aligned}
|f(A_{i,j}) - f(B_{i,j})| &\leq f(A_{i,j}) \cdot (|A_{i,j} - B_{i,j}| + |A_{i,j} - B_{i,j}|^2) \\
&\leq f(A_{i,j}) \cdot |A_{i,j} - B_{i,j}| \cdot (1 + |A_{i,j} - B_{i,j}|) \\
&\leq f(A_{i,j}) \cdot |A_{i,j} - B_{i,j}| \cdot (1 + (2 + \epsilon)r) \\
&\leq f(A_{i,j}) \cdot (2 + \epsilon)r \cdot (1 + (2 + \epsilon)r) \\
&= f(A_{i,j}) \cdot (2 + \epsilon + (2 + \epsilon)^2 r)r \\
&\leq f(A_{i,j}) \cdot (2 + 2\epsilon + 4r)r
\end{aligned}$$

where the 1st step is the result of Fact B.2, the 2nd step follows from straightforward algebraic manipulations, the 3rd step is a consequence of Eq.(4), the 4th step is a consequence of Eq.(4), the 5th step follows from algebraic manipulations, and the 6th step is a result of satisfying **Requirement 3** in the Lemma statement.

**Proof of Part 2.** Similarly, we can prove it.

$\square$

## C.2 Error Control for Normalization

This section focuses on the perturbation of diagonal normalization matrices, which is another important aspect of our error control approach.

**Lemma C.3** (Error Control for Normalization, A general version Lemma 3.3 in [DMS23]). *If the following condition holds*

- **Requirement 1.** *We define $f$ as Definition 4.1.*

- **Requirement 2.** *We define $\mathsf{D}$ as Definition 4.2.*

- **Requirement 3.** $\forall(i,j) \in [n] \times [n]$, *we have* $|f(A_{i,j}) - f(B_{i,j})| \leq f(A_{i,j}) \cdot c_0 r$.

- **Requirement 4.** $\forall(i,j) \in [n] \times [n]$, *we have* $f(A_{i,j}) - f(B_{i,j})| \leq f(B_{i,j}) \cdot c_0 r$.

*Then, it follows that,*

- **Part 1.**

$$|\mathsf{D}(A)_{i,i} - \mathsf{D}(B)_{i,i}| \leq \mathsf{D}(A)_{i,i} \cdot c_0 r \quad \forall i \in [n]$$

- **Part 2.**

$$|\mathsf{D}(A)_{i,i} - \mathsf{D}(B)_{i,i}| \leq \mathsf{D}(B)_{i,i} \cdot c_0 r \quad \forall i \in [n]$$

*Proof.* **Proof of Part 1.** From the above conditions in the lemma statement, it follows that

$$
\begin{aligned}
|\mathsf{D}(A)_{i,i} - \mathsf{D}(B)_{i,i}| &= |(f(A_{i,*}) - f(B_{i,*})) \cdot \mathbf{1}_n| \\
&= |\sum_{j=1}^{n} (f(A_{i,j}) - f(B_{i,j}))| \\
&\leq \sum_{j=1}^{n} |f(A_{i,j}) - f(B_{i,j})| \\
&\leq \sum_{j=1}^{n} f(A_{i,j}) \cdot c_0 r \\
&= f(A_{i,*}) \mathbf{1}_n \cdot c_0 r \\
&= \mathsf{D}(A)_{i,i} \cdot c_0 r
\end{aligned}
$$

where the 1st step follows from algebraic manipulations, the 2nd step is due to algebraic manipulations, the 3rd step is the result of triangle inequality, the 4th step is based on **Requirement 2** in Lemma statement, the 5th step comes from algebraic manipulations and the last step is the result of algebraic manipulations.

**Proof of Part 2.**

The proof is similar to Part 1. So we omit the details here.

$\square$

## C.3 Error of Attention Matrix

In this section, we analyze the error in the attention matrix computation that arises from the perturbations of psd and diagonal normalization matrices.

**Lemma C.4** (A general version of Lemma 3.4 in [DMS23]). *Let $c_1 > 0$ and $c_2 > 0$. If all of the following requirements are met*

- **Requirement 1.** *We define $f$ as Definition 4.1.*

- **Requirement 2.** *We define $\mathsf{D}$ as Definition 4.2.*

- **Requirement 3.**

$$| \mathsf{D}(A)_{i,i} - \mathsf{D}(B)_{i,i} | \leq c_1 \cdot r \cdot \min\{\mathsf{D}(A)_{i,i}, \mathsf{D}(B)_{i,i}\} \ \ \forall i \in [n],$$

- **Requirement 4.**

$$|f(A_{i,j}) - f(B_{i,j})| \leq c_2 \cdot r \cdot \min\{f(A_{i,j}), f(B_{i,j})\} \ \ \forall i, j \in [n] \times [n]$$

*It follows that*

$$\| \mathsf{D}(A)^{-1} f(A) - \mathsf{D}(B)^{-1} f(B) \|_\infty \leq (c_1 + c_2) \cdot r.$$

*Proof.* We first decompose the difference into

$$
\begin{aligned}
& \| \mathsf{D}(A)^{-1} f(A) - \mathsf{D}(B)^{-1} f(B) \|_\infty \\
\leq{}& \| \mathsf{D}(A)^{-1} f(A) - \mathsf{D}(B)^{-1} f(B) \|_\infty + \| \mathsf{D}(B)^{-1} f(B) - \mathsf{D}(B)^{-1} f(B) \|_\infty \\
={}& Z_1 + Z_2
\end{aligned}
$$

where last step is obtained by

$$Z_1 := \| \mathsf{D}(B)^{-1} f(B) - \mathsf{D}(B)^{-1} f(B) \|_\infty,$$

and

$$Z_2 := \| \mathsf{D}(A)^{-1} f(A) - \mathsf{D}(B)^{-1} f(B) \|_\infty.$$

We will present the proof in two parts.

**The first term.** $\forall (i, j) \in [n] \times [n]$, it follows that

$$
\begin{aligned}
Z_1 &= |(\mathsf{D}(A)^{-1} f(A) - \mathsf{D}(B)^{-1} f(B))_{i,j}| \\
&= | \mathsf{D}(A)_{i,i}^{-1} \cdot (f(A)_{i,j} - f(B)_{i,j})| \\
&\leq \mathsf{D}(A)_{i,i}^{-1} \cdot |f(A)_{i,j} - f(B)_{i,j}| \\
&\leq \mathsf{D}(A)_{i,i}^{-1} \cdot c_2 \cdot r \cdot f(A)_{i,j} \\
&\leq c_2 r \cdot (\mathsf{D}(A)^{-1} f(A))_{i,j} \\
&\leq c_2 r,
\end{aligned}
$$

where the 1st step comes from definition, the 2nd step is the result of algebraic manipulations, the 3rd step comes from triangle inequality, the 4th step is based on **Requirement 4** in the lemma statement, the 5th step is the result of algebraic manipulations, and the last step is according to the definition of D.

**The second term.** $\forall (i, j) \in [n] \times [n]$, it follows that

$$
\begin{aligned}
Z_2 &= |(\mathsf{D}(B)^{-1} f(B) - \mathsf{D}(B)^{-1} f(B))_{i,j}| \\
&= |(\mathsf{D}(A)_{i,i}^{-1} - \mathsf{D}(A)_{i,i}^{-1}) f(B)_{i,j}| \\
&= |\frac{\mathsf{D}(A)_{i,i} - \mathsf{D}(B)_{i,i}}{\mathsf{D}(A)_{i,i} \mathsf{D}(B)_{i,i}} f(B)_{i,j}| \\
&\leq |\frac{\mathsf{D}(A)_{i,i} - \mathsf{D}(B)_{i,i}}{\mathsf{D}(A)_{i,i} \mathsf{D}(B)_{i,i}}| \cdot |f(B)_{i,j}| \\
&\leq |\frac{c_1 r \, \mathsf{D}(A)_{i,i}}{\mathsf{D}(A)_{i,i} \mathsf{D}(B)_{i,i}}| \cdot |f(B)_{i,j}| \\
&= c_1 r \cdot | \mathsf{D}(B)_{i,i}^{-1}| \cdot |f(B)_{i,j}|
\end{aligned}
$$

where the 1st step based on definition, the 2nd steps follow from algebraic manipulations, the 3rd step is the result of algebraic manipulations, the 4th step is due to triangle inequality, the 5th step is due to **Requirement 3** in the lemma statement, the last step is due to algebraic manipulations.

Then we have

$$Z_2 = c_1 r \cdot |\, \mathsf{D}(B)_{i,i}^{-1}| \cdot |f(B)_{i,j}|$$
$$= c_1 r \cdot |\, \mathsf{D}(B)_{i,i}^{-1} f(B)_{i,j}|$$
$$= c_1 r \cdot (\mathsf{D}(B)^{-1} f(B))_{i,j}$$
$$\leq c_1 r$$

where the 1st step is the result of the above equation, the 2nd step is due to all the entries are positive, the 3rd step is due to algebraic manipulations and the last step is due to definition of D.

Based on the above deduction, it follows that

$$\|\, \mathsf{D}(A)^{-1} f(A) - \mathsf{D}(B)^{-1} f(B)\|_\infty \leq Z_1 + Z_2$$
$$\leq (c_1 + c_2) r.$$

Thus we complete the proof. $\qquad\square$

## C.4  Error Control

The main result of Section C is presented in this section.

**Theorem C.5** (Formal version of Theorem 4.3). *If all of the following requirements are met*

- *Let $\epsilon \in (0, 0.1)$*

- *Let $r \in (0, 0.1)$*

- *$\|A\|_\infty \leq r$*

- *$(1 - \epsilon)B \preceq A \preceq (1 + \epsilon)B$*

- *We define* D *Definition 4.2.*

- *We define $f$ as Definition 4.1.*

*It follows that*

$$\|\, \mathsf{D}(A)^{-1} f(A) - \mathsf{D}(B)^{-1} f(B)\|_\infty \leq 4 \cdot (1 + \epsilon + 2r) \cdot r$$

*Proof of Theorem 4.3.* By Lemma C.1 and $(1 - \epsilon)B \preceq A \preceq (1 + \epsilon)B$, we have
$$B_{i,j} \in [-(1 + \epsilon)r, (1 + \epsilon)r]. \tag{5}$$
.

By Lemma C.2 and Eq. (5), it follows that

- **Part 1.**
$$|f(A_{i,j}) - f(B_{i,j})| \leq f(A_{i,j}) \cdot (2 + 2\epsilon + 4r) \cdot r \;\; \forall (i, j) \in [n] \times [n].$$

- **Part 2.**
$$|f(A_{i,j}) - f(B_{i,j})| \leq f(B_{i,j}) \cdot (2 + 2\epsilon + 4r) \cdot r \;\; \forall (i, j) \in [n] \times [n].$$

According to the discussion above and using Lemma C.3, we have

- **Part 1.**
$$|\, \mathsf{D}(A)_{i,i} - \mathsf{D}(B)_{i,i}| \leq \mathsf{D}(A)_{i,i} \cdot c_0 r \;\; \forall i \in [n]$$

- **Part 2.**
$$|\, \mathsf{D}(A)_{i,i} - \mathsf{D}(B)_{i,i}| \leq \mathsf{D}(B)_{i,i} \cdot c_0 r \;\; \forall i \in [n]$$

And then by using Lemma C.4, $c_1 = (2 + 2\epsilon + 4r)$ and $c_2 = (2 + 2\epsilon + 4r)$, we have
$$\|\, \mathsf{D}(A)^{-1} f(A) - \mathsf{D}(B)^{-1} f(B)\|_\infty \leq 4 \cdot (1 + \epsilon + 2r) \cdot r$$

$\qquad\square$

# D   Analysis of Gaussian Sampling Mechanism

We denote the output of our privacy algorithm as $Z$. In Section D.1, we present the computation tools that we use to implement our approach. In Section D.2, we perform spectral decomposition of $A := \mathcal{M}(\mathcal{Y})^{1/2}\mathcal{M}(\mathcal{Y}')^{-1}\mathcal{M}(\mathcal{Y})^{1/2}$ and derive some important conclusions from it. Then, in Section D.3, we transform $Z$ into a format that is based on the spectral decomposition of $A$. In Section D.4, We present the upper bound of $\mathbb{E}[Z]$, which is useful in the following section. In Section D.5, we demonstrate that $Z$ is sub-exponential, which allows us to control the upper bound of $\Pr[Z \geq \epsilon]$ where $\epsilon \in (0, 1)$. Finally, we present our main result in Section D.6, which is that our Algorithm 2 is differential privacy.

## D.1   Computation Tools

This section is dedicated to presenting the computational tools that we use to implement our approach.

**Definition D.1.** *We define* $\Sigma_1, \Sigma_2$ *as Definition 4.7. Let us define*

- $A := \Sigma_1^{1/2}\Sigma_2^{-1}\Sigma_1^{1/2}$

- $B := \Sigma_2^{1/2}\Sigma_1^{-1}\Sigma_2^{-1/2}$

- $C := \Sigma_1^{-1/2}\Sigma_2^{1/2}$

**Lemma D.2.** *Let* $A, B$ *and* $C$ *be defined as Definition D.1. Then we have*

- **Part 1.** $A^{-1} = CC^\top$.

- **Part 2.** $B = C^\top C$.

- **Part 3.** $A^{-1}, B$ *have the same eigenvalue.*

*Proof.* Note that $\Sigma_1$ and $\Sigma_2$ are symmetric, we can easily have the proof as follows.

**Proof of Part 1.**

$$
\begin{aligned}
A^{-1} &= (\Sigma_1^{1/2}\Sigma_2^{-1}\Sigma_1^{1/2})^{-1} \\
&= (\Sigma_1^{1/2}\Sigma_2^{-1/2}\Sigma_2^{-1/2}\Sigma_1^{1/2})^{-1} \\
&= (\Sigma_2^{-1/2}\Sigma_1^{1/2})^{-1}(\Sigma_1^{1/2}\Sigma_2^{-1/2})^{-1} \\
&= (\Sigma_1^{1/2}\Sigma_2^{-1/2})(\Sigma_2^{-1/2}\Sigma_1^{1/2}) \\
&= CC^\top
\end{aligned}
\tag{6}
$$

**Proof of Part 2.**

$$
\begin{aligned}
B &= \Sigma_2^{-1/2}\Sigma_1\Sigma_2^{-1/2} \\
&= (\Sigma_2^{-1/2}\Sigma_1^{1/2})(\Sigma_1^{1/2}\Sigma_2^{-1/2}) \\
&= C^T C
\end{aligned}
\tag{7}
$$

**Proof of Part 3.**   It simply follows from Eq.(6) and Eq.(7).   $\square$

## D.2   Spectral Decomposition

This section is focused on the spectral decomposition of $A$, which we perform to gain insights into its properties. By analyzing the spectral decomposition, we are able to draw important conclusions about $A$ that are relevant to our approach.

**Lemma D.3.** *If all of the following requirements are met*

- **Requirement 1.** *We define* $A$ *as Definition D.1.*

- **Requirement 2.** *Let $\lambda_1 \cdots \lambda_n$ be eigenvalues of $A$.*

- **Requirement 3.** *Let $A = \sum_{j=1}^n \lambda_j v_j v_j^\top$ be spectral decomposition for $A$.*

- **Requirement 4.** *Let $\Delta$ be denoted as Definition 4.5.*

- **Requirement 5.** *Let $M, \mathcal{M}$ be denoted as Definition 4.4 and $M \leq \Delta$.*

*We have*

- $\sum_{j=1}^n (\lambda_j - 1)^2 \leq \Delta^2$.

- $\sum_{j=1}^n (1 - \frac{1}{\lambda_j})^2 \leq \Delta^2$.

*Proof.* we have

$$\sum_{j=1}^n (\lambda_j - 1)^2 = \|A - I\|_F^2$$
$$\leq \Delta^2$$

where the 1st step is based on **Requirement 3** in the lemma statement and the last step is due to **Requirement 5** in lemma statement.

Similarly, we have

$$\sum_{j=1}^n (1 - \frac{1}{\lambda_j})^2 = \|I - A^{-1}\|_F^2$$
$$= \|I - B\|_F^2$$
$$\leq \Delta^2$$

where the 1st step is due to **Requirement 3** in the lemma statement, the 2nd step follows from swapping the roles of $\mathcal{Y}, \mathcal{Y}'$ and the last step is due to Lemma D.2. $\qquad\square$

### D.3 The Transformation for Output

In Section D.3, we describe the process of transforming the output $Z$ of our privacy algorithm into a format that is based on the spectral decomposition of $A$.

**Lemma D.4.** *If all of the following requirements are met*

- **Requirement 1.***We define $Z$ and $h_{i,j}$ as Definition 4.8.*

- **Requirement 2.** *Let $A$ be denoted as Definition D.1.*

- **Requirement 3.** *Let $\lambda_1, \cdots, \lambda_n$ demote the eigenvalue of $A$.*

*Then we have*

$$Z = \frac{1}{2} \sum_{i=1}^k \sum_{j=1}^n \left( (\lambda_j - 1)h_{i,j}^2 - \log(\lambda_j) \right)$$

*Proof.* The privacy loss random variable $Z$ can be expressed as follows:

$$Z = \sum_{i=1}^k \log \left( \frac{\det(\Sigma_1)^{-\frac{1}{2}} \exp(-\frac{1}{2} g_i^\top \Sigma_1^{-1} g_i)}{\det(\Sigma_2)^{-\frac{1}{2}} \exp(-\frac{1}{2} g_i^\top \Sigma_2^{-1} g_i)} \right)$$
$$= \sum_{i=1}^k \left( \frac{1}{2} g_i^\top (\Sigma_2^{-1} - \Sigma_1^{-1}) g_i - \frac{1}{2} \log \left( \frac{\det(\Sigma_1)}{\det(\Sigma_2)} \right) \right)$$
$$= \frac{1}{2} \sum_{i=1}^k \left( \left( \Sigma_1^{-1/2} g_i \right)^\top (A - I) \left( \Sigma_1^{-1/2} g_i \right) - \log \det(A) \right)$$

$$= \frac{1}{2} \sum_{i=1}^{k} \sum_{j=1}^{n} \left( (\lambda_j - 1)h_{i,j}^2 - \log(\lambda_j) \right)$$

where the 1st step is based on **Requirement 1** in the lemma statement, the 2nd step follows from rearranging the terms, the 3rd step is based on **Requirement 2** in the lemma statement, and the last step is by taking the spectral decomposition of $A$. □

## D.4 The Upper Bound for Expectation

In Section D.4, we provide an upper bound on the expected value of $Z$, which is a useful result for the subsequent section.

**Lemma D.5.** *If all of the following requirements are met*

- **Requirement 1** *We define $Z$ as Definition 4.8.*

- **Requirement 2** *Let $\epsilon \in (0, 1)$ and $k \in \mathbb{N}$.*

- **Requirement 3.** *Let $A$ be denoted as Definition D.1.*

- **Requirement 4.** *Let $\lambda_1, \cdots, \lambda_n$ denote the eigenvalue of $A$.*

- **Requirement 5.** *Let $\Delta$ be denoted as Definition 4.5.*

- **Requirement 6.** *Let $M, \mathcal{M}$ be denoted as Definition 4.4 and $M \leq \Delta$.*

*we have*

$$\mathbb{E}[Z] \leq \frac{\epsilon}{2}$$

*Proof.*

$$
\begin{aligned}
\mathbb{E}[Z] &= \frac{k}{2} \sum_{j=1}^{n} (\lambda_j - 1 - \log(\lambda_j)) \\
&\leq \frac{k}{2} \sum_{j=1}^{n} (\lambda_j - 2 + \frac{1}{\lambda_j}) \\
&= \frac{k}{2} \sum_{j=1}^{n} (\lambda_j - 1)(1 - \frac{1}{\lambda_j}) \\
&\leq \|A - I\|_F \cdot \|I - A^{-1}\|_F \\
&\leq \frac{k}{2} \Delta^2 \\
&\leq \frac{\epsilon}{2}
\end{aligned}
$$

where the 1st step follows from linearity of expectation and Lemma D.4, the 2nd step is the result of $\lambda_j > 0$ and $\log(x) > 1 - \frac{1}{x}$ for $x > 0$, the 3rd step follows from simple factorization, the fourth step follows from Cauchy-Schwarz, the fifth step follows from Lemma D.3 and **Requirement 6** in the lemma statement, and the last step follows from $\Delta < \frac{\epsilon}{\sqrt{k}}$ and $\epsilon < 1$. □

## D.5 Sub-Exponential

In Section D.5, evidence is provided that supports the claim that $Z$ is sub-exponential. This is significant because it enables us to limit the maximum probability of the event $Z \geq \epsilon$, which is crucial in ensuring differential privacy.

**Lemma D.6.** *If all of the following requirements are met*

- **Requirement 1.***We define $Z$ as Definition 4.8.*

- **Requirement 2.** *Let $\epsilon \in (0,1)$ and $\delta \in (0,1)$.*
- **Requirement 3.** *Let $\Delta$ be denoted as Definition 4.5 and $\Delta < 1$.*
- **Requirement 4.** *Let $M, \mathcal{M}$ be denoted as Definition 4.4 and $M \le \Delta$.*
- **Requirement 5.** $k \in \mathbb{N}$.

*we have*

$$\Pr[Z > \epsilon] \le \delta$$

*Proof.* First, we will prove $Z$ is sub-exponential.

**Proof of Sub Exponential** Let $A$ be dented as Definition D.1 and $h_{i,j}$ be denoted as Definition 4.8. Since $h_{i,j} \sim \chi_1^2$, Lemma B.7 and Lemma B.6, we can say $Z$ is sub-exponential with

- $\nu = \sqrt{k}\|I - A\|_F$
- $\alpha = 2\|I - A\|_F$

By Lemma D.3, we have

- $\nu = \sqrt{k}\|A - I\|_F \le \sqrt{k}\Delta$
- $\alpha = 2\|A - I\|_F \le 2\Delta$

**Proof of Upper Bound for $\mathbb{E}[Z]$.** Under **Requirement 3** and **Requirement 4**, by using Lemma D.5, we have

$$\mathbb{E}[Z] \le \epsilon/2 \tag{8}$$

**Proof of Upper Bound** By using Lemma B.5 (sub-exponential tail bound), we have

$$
\begin{aligned}
\Pr[Z > \epsilon] &< \Pr[Z - \mathbb{E}[Z] > \epsilon/2] \\
&\le \max\left\{ \exp(-\frac{(\epsilon/2)^2}{2\nu^2}), \exp(-\frac{\epsilon/2}{2\alpha}) \right\} \\
&\le \delta
\end{aligned}
$$

where the 1st step is the reuslt of Eq. (8), the 2nd step is the reuslt of Lemma B.5, and the last step follows from **Requirement 3** in the lemma statement. $\square$

## D.6 Analysis of Gaussian Sampling

This section contains our main result in Section D, which we present as follows. The following theorem statement can be viewed as a variation of Theorem 5.1 in [AKT+22].

**Theorem D.7** (Formal version of Theorem 4.9, Analysis of the Gaussian Sampling Mechanism ). *If all of the following requirements are met*

- **Requirement 1.** *Let $\epsilon \in (0,1)$ and $\delta \in (0,1)$.*
- **Requirement 2.** $k \in \mathbb{N}$.
- **Requirement 3.** *Let $\Delta$ be denoted as Definition 4.5 and $\Delta < 1$.*
- **Requirement 4.** *Let $M, \mathcal{M}$ be denoted as Definition 4.4 and $M \le \Delta$.*
- **Requirement 5.** *An input $\Sigma = \mathcal{M}(\mathcal{Y})$.*
- **Requirement 6.** $\rho = O(\sqrt{(n^2 + \log(1/\gamma))/k} + (n^2 + \log(1/\gamma))/k)$.

*Then, there exists an algorithm 2 such that*

- *Part 1. Algorithm 2 is $(\epsilon, \delta)$-DP (with respect to the original dataset $\mathcal{Y}$).*

- *Part 2. outputs $\widehat{\Sigma} \in \mathbb{S}_+^n$ such that with probabilities at least $1 - \gamma$,*

$$\|\Sigma^{-1/2}\widehat{\Sigma}\Sigma^{-1/2} - I_n\|_F \leq \rho$$

- *Part 3.*

$$(1 - \rho)\Sigma \preceq \widehat{\Sigma} \preceq (1 + \rho)\Sigma$$

*Proof.* We denote $Z$ as Definition 4.8 which is as the output of algorithm 2. The utility guarantee is immediately implied by Theorem B.3. We then focus on the proof of privacy. By Lemma D.6, we have

$$\Pr[Z > \epsilon] \leq \delta \tag{9}$$

And then by Theorem B.4 and Eq. (9), Algorithm 2 is proved as $(\epsilon, \delta)$-differential private.

**Proof of Part 3.**

$$\|\Sigma^{-1/2}\widehat{\Sigma}\Sigma^{-1/2} - I_n\| \leq \|\Sigma^{-1/2}\widehat{\Sigma}\Sigma^{-1/2} - I_n\|_F$$
$$\leq \rho$$

Thus,

$$(1 - \rho)I_n \preceq \Sigma^{-1/2}\widehat{\Sigma}\Sigma^{-1/2} \preceq (1 + \rho)I_n$$

which is equivalent to

$$(1 - \rho)\Sigma \preceq \widehat{\Sigma} \preceq (1 + \rho)\Sigma$$

$\square$

# E    More Sensitivity Lemma

In this section, we provide more lemmas related to sensitivity.

**Lemma E.1** (Formal version of Lemma 4.10)**.** *If $X \in \mathbb{R}^{n \times d}$ and $\widetilde{X} \in \mathbb{R}^{n \times d}$ are neighboring dataset (see Definition 1.5 and Definition 1.6), then $(1 - 2\alpha\beta/\eta)XX^\top \preceq \widetilde{X}\widetilde{X}^\top \preceq (1 + 2\alpha\beta/\eta)XX^\top$.*

*Proof.* Let $i \in [d]$ be index that $X_{*,i}$ and $\widetilde{X}_{*,i}$ are different (See Definition 1.6).

We have

$$\begin{aligned}
\widetilde{X}\widetilde{X}^\top &= \sum_{j=1}^d \widetilde{X}_{*,j}\widetilde{X}_{*,j}^\top \\
&= (\sum_{j \in [d]\backslash\{i\}} \widetilde{X}_{*,j}\widetilde{X}_{*,j}^\top) + \widetilde{X}_{*,i}\widetilde{X}_{*,i}^\top \\
&= (\sum_{j \in [d]\backslash\{i\}} X_{*,j}X_{*,j}^\top) + \widetilde{X}_{*,i}\widetilde{X}_{*,i}^\top \\
&= XX^\top - X_{*,i}X_{*,i}^\top + \widetilde{X}_{*,i}\widetilde{X}_{*,i}
\end{aligned}$$

where the first step is the result of matrix multiplication, the second step is from simple algebra, the third step follows from Definition 1.6, and the last step comes from simple algebra.

We know that

$$\begin{aligned}
\|X_{*,i}X_{*,i}^\top - \widetilde{X}_{*,i}\widetilde{X}_{*,i}\| &= \|X_{*,i}X_{*,i}^\top - X_{*,i}\widetilde{X}_{*,i}^\top + X_{*,i}\widetilde{X}_{*,i}^\top - \widetilde{X}_{*,i}\widetilde{X}_{*,i}\| \\
&\leq \|X_{*,i}X_{*,i}^\top - X_{*,i}\widetilde{X}_{*,i}^\top\| + \|X_{*,i}\widetilde{X}_{*,i}^\top - \widetilde{X}_{*,i}\widetilde{X}_{*,i}\|
\end{aligned}$$

$$\leq \|X_{*,i}\|_2 \cdot \|X_{*,i} - \widetilde{X}_{*,i}\|_2 + \|X_{*,i} - \widetilde{X}_{*,i}\|_2 \cdot \|\widetilde{X}_{*,i}\|_2$$
$$\leq 2\alpha\beta \tag{10}$$

where the first step is from adding a new term $X_{*,i}\widetilde{X}_{*,i}^\top$, the second step follows from the triangle inequality, the third step follows from Fact B.1, and the last step is due to Definition 1.5 and Definition 1.6.

Thus, we have $\widetilde{X}\widetilde{X}^\top \succeq XX^\top - 2\alpha\beta I_n \succeq (1 - 2\alpha\beta/\eta)XX^\top$. Similarly, we have $\widetilde{X}\widetilde{X}^\top \preceq XX^\top + 2\alpha\beta I_n \preceq (1 + 2\alpha\beta/\eta)XX^\top$. $\qquad\square$

The following is the presentation of the additional sensitivity lemma, which further extends the conclusion of Lemma 4.10 in Section 4.3. We use the following lemma in the proof of our main result, Theorem 3.1, presented in Section 3.

**Lemma E.2.** *If the following conditions hold*

- *Let $\alpha$ and $\eta$ be defined in Definition 1.5.*

- *Let $\beta$ be defined in Definition 1.6.*

- *$X$ and $\widetilde{X}$ are neighboring datasets such that*

$$(1 - 2\alpha\beta/\eta)XX^\top \preceq \widetilde{X}\widetilde{X}^\top \preceq (1 + 2\alpha\beta/\eta)XX^\top$$

*Then, we have*

- *Part 1.*

$$\|(XX^\top)^{-1/2}\widetilde{X}\widetilde{X}^\top(XX^\top)^{-1/2} - I\| \leq 2\alpha\beta/\eta$$

- *Part 2.*

$$\|(XX^\top)^{-1/2}\widetilde{X}\widetilde{X}^\top(XX^\top)^{-1/2} - I\|_F \leq 2\sqrt{n}\alpha\beta/\eta$$

*Proof.* The proof is straightforward, and we omit the details here. $\qquad\square$

