# OpenReview forum: "Differentially Private Attention Computation"
_NeurIPS.cc/2024/Workshop/SafeGenAi — SafeGenAi Poster_

### Official Review · Reviewer_uTAm · 2024-10-09
**The experiments are missing.**

**Rating:** 4
**Confidence:** 5

**Review:**

### Paper Summary

This paper proposes a novel algorithm for approximating the attention matrix while ensuring differential privacy (DP). The work builds on recent advancements in fast attention computation and differential privacy matrix publishing. The authors discuss the privacy and security issues associated with large language models (LLMs) when handling sensitive information and improve the computation of the attention mechanism through static computation techniques, aiming to enhance computational efficiency and reduce the risk of sensitive information leakage.

### Strengths

1. **Innovative**: The paper introduces a new method for differential privacy in attention calculation, which is an important addition to existing privacy protection technologies for language models.
2. **Practical Value**: Given the extensive application of LLMs across various fields, providing a privacy-preserving method for attention calculation holds significant practical value.
3. **Theoretical and Practical Integration**: The paper not only theoretically defines the models for differential privacy and attention calculation but also explores their practical applications, especially in static computation settings.

### Weaknesses

1. **Lack of Experimental Validation**: The paper does not provide experimental data to demonstrate the effectiveness and efficiency of the proposed method, and it lacks comparisons with existing technologies.
2. **Insufficient Security Analysis**: Although privacy protection is discussed, there is a lack of comprehensive analysis of potential security threats, such as model inversion attacks.
3. **Main Result Clarity**: It is suggested that the authors more clearly articulate the benefits brought by the proposed method and address real world problems, ideally providing practical examples.

### Suggestions for Improvement

1. **Add Experimental Validation**: The authors should consider including experimental sections that demonstrate the advantages and disadvantages of the proposed algorithm through comparisons with traditional attention mechanisms and other privacy-preserving methods.
2. **Detailed Security Assessment**: It is recommended to add an assessment of the potential security threats faced by the algorithm, particularly measures to protect against data breaches and model attacks.
3. **Expand Discussion and Application Scenarios**: The authors could consider expanding the discussion on the algorithm's performance in different application scenarios, such as its performance on different sizes of datasets and its feasibility in real-time systems.

If the authors add more experimental evidence demonstrating the effectiveness of their method, I would consider changing my score, looking forward to more work from the authors.

---

### Official Review · Reviewer_WJB9 · 2024-10-09
**Differentially Private Attention Computation**

**Rating:** 5
**Confidence:** 2

**Review:**

The paper proposes a differentially private algorithm for approximating the attention matrix of large language models. The authors focus on static computation for attention, where the attention weights between the encoder and decoder are computed only once and reused during decoding. Moreover, the paper details the notation used throughout the paper. Additionally, the paper conducts several analyses such as perturbations in the attention computation and the Gaussian sampling mechanism. However, the paper lists many definitions and theorems but does not spend adequate space explaining them, which makes the paper difficult to follow and comprehend.

---

### Official Review · Reviewer_vWJp · 2024-10-09
**The paper proposes a promising and novel method in which they approximate the attention matrix while providing differential privacy guarantees.**

**Rating:** 6
**Confidence:** 2

**Review:**

Strengths: Providing an approach in which the attention computations are differentially private is important, especially in the context of handling private/sensitive data. The authors of the paper present a mathematical framework for this computation. It includes strong theoretical proofs as well as error analyses.

Weaknesses:
1. The paper would benefit from having a discussion on how restrictive the assumptions being made on the data are and how and if this differs from real-world data.
2. The paper should also consider addressing the scalability of this approach. Please provide complexity analysis and/or discuss scalability on very large datasets (often used to build LLMs).
3. The authors should consider doing an empirical validation of their findings on real-world data to improve the robustness/practicality of their theoretical findings.

---

### Official Review · Reviewer_VuRz · 2024-10-09
**approximating the attention matrix**

**Rating:** 6
**Confidence:** 4

**Review:**

This paper presents a novel approach for differentially private attention computation, leveraging advancements in fast attention mechanisms and private matrix publishing. The authors rigorously derive privacy guarantees and error bounds for approximating the attention matrix, demonstrating that their method achieves efficient approximation with (ϵ, δ)-differential privacy. The theoretical analysis is thorough, with clear definitions and logical flow.